# *Pneumocystis jirovecii* Colonization in Mexican Patients with Chronic Obstructive Pulmonary Disease

**DOI:** 10.3390/tropicalmed8030137

**Published:** 2023-02-24

**Authors:** Marcela Plascencia-Cruz, Arturo Plascencia-Hernández, Yaxsier De Armas-Rodríguez, Gabino Cervantes-Guevara, Guillermo Alonso Cervantes-Cardona, Sol Ramírez-Ochoa, Alejandro González-Ojeda, Clotilde Fuentes-Orozco, Francisco Javier Hernández-Mora, Carlos Miguel González-Valencia, Andrea Pérez de Acha-Chávez, Enrique Cervantes-Pérez

**Affiliations:** 1Department of Internal Medicine, Hospital Civil de Guadalajara “Fray Antonio Alcalde”, Guadalajara 44280, Jalisco, Mexico; 2Health Sciences University Center, Universidad de Guadalajara, Guadalajara 44100, Jalisco, Mexico; 3Department of Pediatric Infectious Diseases, Hospital Civil de Guadalajara “Fray Antonio Alcalde”, Guadalajara 44280, Jalisco, Mexico; 4Molecular Biology Laboratory, Instituto de Medicina Tropical “Pedro Kourí”, La Habana 11400, Cuba; 5Department of Welfare and Sustainable Development, Centro Universitario del Norte, Universidad de Guadalajara, Colotlán 46200, Jalisco, Mexico; 6Department of Gastroenterology, Hospital Civil de Guadalajara “Fray Antonio Alcalde”, Guadalajara 44280, Jalisco, Mexico; 7Biomedical Research Unit 02, Specialties Hospital of the Western National Medical Center, Mexican Institute of Social Security, Guadalajara 44329, Jalisco, Mexico; 8Human Reproduction, Growth and Child Development Clinic, Health Sciences University Center, Universidad de Guadalajara, Guadalajara 44100, Jalisco, Mexico; 9Department of Research Ethics, Hospital Hispano, Guadalajara 44140, Jalisco, Mexico; 10Department of Geriatrics, Instituto Nacional de Ciencias Médicas y Nutrición Salvador Zubirán, Mexico City 14080, Mexico

**Keywords:** *Pneumocystis jirovecii* pneumonia (PcP), chronic obstructive pulmonary disease (COPD), colonization, nested PCR, oropharyngeal wash (OPW)

## Abstract

The prevalence of colonization by *Pneumocystis jirovecii* (*P. jirovecii*) has not been studied in Mexico. We aimed to determine the prevalence of colonization by *P. jirovecii* using molecular detection in a population of Mexican patients with chronic obstructive pulmonary disease (COPD) and describe their clinical and sociodemographic profiles. We enrolled patients discharged from our hospital diagnosed with COPD and without pneumonia (*n* = 15). The primary outcome of this study was *P. jirovecii* colonization at the time of discharge, as detected by nested polymerase chain reaction (PCR) of oropharyngeal wash samples. The calculated prevalence of colonization for our study group was 26.66%. There were no statistically significant differences between COPD patients with and without colonization in our groups. Colonization of *P. jirovecii* in patients with COPD is frequent in the Mexican population; the clinical significance, if any, remains to be determined. Oropharyngeal wash and nested PCR are excellent cost-effective options to simplify sample collection and detection in developing countries and can be used for further studies.

## 1. Introduction

The risks and prevalence of colonization by *Pneumocystis jirovecii* (*P. jirovecii)* in patients with chronic obstructive pulmonary disease (COPD) have been studied in the past; however, there is no consensus regarding the risk of colonization, and the prevalence in different populations is widely variable (ranging from 16 to 55%) [1,2,3]. Nested polymerase chain reaction (PCR) is a cost-effective modification of the standard PCR that serves to increase the sensitivity and specificity of the test even when the level of DNA in a sample is low, making this technique a good method for detection that has been used in previous studies [2,3,4,5,6].

Only a few reports have described a possible association with colonization by *P. jirovecii* and a rise in disease severity and/or inflammatory response in patients with COPD without human immunodeficiency virus (HIV) infection [5,6,7,8,9]. The relationship between COPD exacerbations and the presence of *P. jirovecii* has been suggested in some studies; however, it has not been fully elucidated [7]. Some hypotheses have been proposed regarding the possible association of *P. jirovecii* with an increase in the inflammatory response and how it may influence the pathophysiology of COPD [8]. Obtaining the prevalence in different populations is a first step toward understanding this phenomenon and investigating its clinical relevance.

To our knowledge, there is no published evidence of the prevalence of colonization by *P. jirovecii* in Mexican patients with COPD.

In this exploratory study, we aimed to determine the prevalence of colonization by *P. jirovecii* in a small population of Mexican patients with COPD and describe their clinical and sociodemographic profiles with the goal of identifying possible associations for future studies.

## 2. Materials and Methods

For this analytical cross-sectional exploratory study, we enrolled individuals discharged from the Unit of Internal Medicine of the Hospital Civil de Guadalajara Fray Antonio Alcalde (HCG), located in Guadalajara, Mexico, between April and December 2019. Patients who were diagnosed with COPD in accordance with Global Initiative for Chronic Obstructive Lung Disease Guidelines [10] without clinical and/or radiological signs of active pneumonia and who were able and willing to give an oropharyngeal wash (OPW) sample were selected for the study. All patients showed HIV rapid test negativity before OPW sample collection and were without HIV infection or acquired immunodeficiency syndrome (AIDS) diagnosis. Patient demographics, clinical, imaging, and laboratory data and outcomes were extracted from medical records.

The primary objective of this study was to use nested PCR to determine the prevalence of colonization by *P. jirovecii* in COPD patients in our study group. To address this objective, oropharyngeal washes were obtained using a well-established protocol [11]. The genetic material of *P. jirovecii* was purified from the OPW sample using the QIAamp DNA reagent kit (Qiagen, Barcelona, Spain) according to the manufacturer’s instructions. The DNA was stored at −20 °C until use. For the amplification of the genetic material of *P. jirovecii*, a 347 base pair (bp) fragment (sequence 5′-3′: GATGGCTGTTTCCAAGCCCAGTGTACGTTGCAAAGTACTC) in the first reaction and a 260 bp fragment (sequence 5′-3′: GTGAAATACAAATCGGACTAGGTCACTTAATATTAATTGGGGAGC) in the second reaction of the gene that codes for the mitochondria large subunit ribosomal ribonucleic acid gene (mt LSU rRNA) were the target sequences [12]. Briefly, 50 μL of reaction mixture was prepared for the first and second rounds of amplification of the *P. jirovecii* DNA, obtained from the OPW sample of the patients. The reaction mixture contained 10 mM Tris/HCl (pH 8.3), 50 mM KCl, 1.5 mM MgCl_2_, 200 μM deoxyribonucleotides (dNTPs), 0.5 μM primers, 2 units of Taq DNA polymerase, and 5 μL of template DNA (Bioline, London, UK). The amplification profile was the same for the first and second rounds: 94 °C for 4 min, 40 cycles of 1 min at 94 °C, 1 min at 55 °C, and 1.5 min at 72 °C, with a final extension of 7 min at 72 °C. The primers used were synthesized by Invitrogen (Life Technologies S.A., Madrid, Spain). The amplification reaction was carried out using a Biometra TGradient thermal cycler (Whatman Biometra, Minden, Germany). A negative control was used for each assay, whereby the reaction mixture contained sterile distilled water instead of template DNA. A positive control (DNA from an OPW sample of an HIV-positive patient with *Pneumocystis jirovecii* pneumonia (PcP), as verified by microscopic staining and PCR methods), was also included in each amplification reaction.

To prevent contamination, pipettes with filters were used for all manipulations. DNA extraction and preparation of the reaction mixture were performed in two different rooms using separate laminar-flow hoods. The PCR procedure and analysis of PCR products were performed in another room. Control samples were performed simultaneously with the OPW samples.

A positive result in the first and second rounds of amplification, with clinical signs and symptoms and/or radiological signs compatible with pneumonia, was considered PcP. A positive result only in the second round, without clinical or radiological signs of PcP, was considered colonization [13].

For the data and statistical analysis, absolute and relative frequencies were calculated for all qualitative variables and are presented as percentages and proportions. Proportions were compared through Fisher’s test, and confidence intervals (CI) were calculated. Quantitative variables were analyzed with measures of central tendency and dispersion (median and standard deviation), and differences between groups were analyzed with Student’s *t*-test. All statistical analyses were performed with GraphPad Prism version 9.3.1.471 (Dotmatics, Boston, MA, USA). All *p* values < 0.05 within a confidence interval of 95% were considered statistically significant.

## 3. Results

Of the 50 patients with a COPD diagnosis in the period of our study, only 15 (30%) were included in accordance with the inclusion criteria. Other COPD patients (n = 35, 70%) were excluded due to the inability to obtain an OPW sample for analysis.

For the first reaction of the one-round PCR, none of the patient samples were positive using the primary reaction primers; however, 4 of the 15 samples were positive in the nested PCR with the second round of primers. The calculated prevalence of colonization by *P. jirovecii* in this group of patients was 26.7%.

Table 1 shows the sociodemographic characteristics together with clinical and biochemical data for the study groups.

All OPW samples were sent to microbiological culture for the isolation of bacteria. *Pseudomonas aeruginosa* was isolated in one (25%) patient with colonization by *P. jirovecii* vs. eight (72.7%) patients without colonization by *P. jirovecii*. In two patients with colonization by *P. jirovecii*, *Streptococcus viridians* and *Neisseria spp*. were also isolated, and *Acinetobacter baumannii* was isolated in two different patients of the same group; none of these bacteria were isolated in patients who were not colonized by *P. jirovecii*.

## 4. Discussion

The prevalence rate of colonization by *P. jirovecii* in COPD patients in this study group was 26.7%. To our knowledge, this is the first study to describe this parameter and the clinical profile of COPD patients colonized with *P. jirovecii* in a Mexican population (albeit a small study group), and further studies with a larger study group and scope are needed. Additionally, the method used for sample collection for nested PCR was OPW, which is a cost-effective and noninvasive method for sample collection and diagnosis [14].

The issue of the colonization of *P. jirovecii* and other fungi has been addressed more frequently in recent years [3], with the advent of molecular techniques that favor more accurate detection; however, as mentioned, the clinical relevance is not yet clear or well understood [15,16,17]. Although the clinical manifestations of *P. jirovecii* infection have so far been associated with severe immunosuppression (especially that associated with AIDS) [11], moderate immunosuppression, which can be caused by multiple factors, including the proinflammatory state secondary to COPD [18], has been associated with predisposition for colonization [18,19]. Colonization plays an important role in the mechanism of *P. jirovecii* transmission [20]. Patients with colonization can be the source of infection in a susceptible population, and they themselves can develop PcP if they have immunosuppression [21,22,23,24]. Studies on animals and humans [12,20] have documented a possible role in the local and systemic inflammatory response accompanied by structural changes and impaired lung function that can be caused by *Pneumocystis* colonization [2,20]. There is still no conclusive evidence of causation or clinical significance [3].

In this study, all samples were obtained through OPW, a noninvasive procedure that can be used to study the prevalence and effects of colonization by *P. jirovecii* in various clinical settings [14]. There are a variety of new molecular techniques that can increase sensitivity and specificity to detect microorganisms, and they can be applied to samples obtained from less-invasive procedures, such as OPW [14,25]. In developing countries, access to various molecular techniques is difficult, and there are often not enough specialized personnel for sampling with invasive procedures, such as bronchoalveolar lavage (BAL). Although BAL is a sensitive technique for the isolation of *P. jirovecii* [21,22,23], especially in the case of colonization due to the low fungal load, noninvasive tests have shown good sensitivity for detection [14], and have the advantage of being able to be performed with greater ease than more invasive tests such as BAL in certain clinical scenarios when the patient’s state of health could prevent sample collection. One of the ways to increase the sensitivity of noninvasive tests is to use two paired tests (such as OPW and, for example, nasal swab), thus minimizing underestimation of *P. jirovecii* colonization [14].

The prevalence of colonization in COPD is quite variable (from 10 to 65%) [3]; this may be due to the techniques used to assess the prevalence, since molecular techniques are much more sensitive than staining [4]. With the extensive use of these molecular techniques, such as nested PCR, we could gain a better idea of the real prevalence of colonization in this patient population [20,26,27].

Interestingly, it has been seen that HIV-infected patients colonized by *P. jirovecii* have up to an 8-fold higher risk of airway obstruction compared to noncolonized patients [9]; however, in non-HIV-infected patients, no similar relationship was found, even though several studies have reported a higher prevalence of colonization in patients with severe COPD compared to those with moderate COPD [28]. Some of these studies have associated this finding with the severity of the disease, intuiting a causal association; however, the evidence in this regard is not conclusive [20,28].

Some animal studies have detected an association between colonization by *P. jirovecii* and impaired lung function [29], while in humans the only causal association, whose clinical implication is still unknown, is the increase in the systemic inflammatory response and increased Th1 activity [8,20]. This inflammatory response could generate a decrease in the lung defense mechanisms and an alteration of the lung microbiome and function [30,31]. In our study, spirometry values and long-term follow-up were unavailable.

We did not find statistically significant differences in our sample of patients, but even though this prevents us from drawing conclusions about possible clinical associations with colonization, it confirms information reported in the international literature about the high prevalence of colonization in patients with COPD [2,3,4]. Hence, future studies with a larger study group that allows us to determine the clinical and epidemiological relevance in our population are needed. If an association between colonization with *P. jirovecii* and impaired lung function could be verified in the future, it would be worth investigating the colonization time necessary for the increased risk of deterioration. For this, we need to take into account the current evidence about the possible cyclical nature of colonization in patients without HIV infection [32,33]. For example, in patients with cystic fibrosis, a constant cycle of clearance and permanence of colonization by *P. jirovecii* was observed with genotypic variation in the strains in each cycle [34].

There are some reports with possible associations with the occurrence of exacerbation or the deterioration of lung function in patients with COPD [3,35,36], but more studies that include a wide variety of populations are still needed in this regard.

The limitations of this study are as follows: the sample size of patients was small, and because of this, it was difficult to establish clinical and statistically significant differences for the variables; pairing OPW with another noninvasive method for sample collection might have increased the sensitivity and given us a better sense of the prevalence in our population [14]; lastly, the nature of this study was exploratory, so a future study would greatly benefit from the establishment of a proper statistical hypothesis and sample size calculation for the study group.

Although this exploratory study has various limitations, it adds more evidence to the data in the scientific literature about the prevalence of colonization of *P. jirovecii* in patients with COPD [3,36], which warrants future research. These findings are relevant to obtaining better knowledge on the current state of the prevalence of *P. jirovecii* in this population of patients and geographical context, its importance as a research topic, and by carrying out studies with a larger number of patients and molecular techniques that even allow for us to detect the microbiological load (such as RT-PCR), and determine its possible clinical relevance and/or role in the cycle of transmission to the immunosuppressed population with a higher risk of presenting PcP [3]. The molecular techniques at our disposal are far more powerful and accessible than the techniques available a few years ago, and procedures such as OPW are cost-effective options for easy sample collection, especially in developing countries.

## Figures and Tables

**Table 1 tropicalmed-08-00137-t001:** Sociodemographic characteristics together with clinical and biochemical data for COPD patients (colonization vs. noncolonization).

	COPD Patients with Colonization (n = 4)	COPD Patients without Colonization (n = 11)	Prevalence Ratio	CI 95%	*p*
Male	3 (75%)	5 (45.5%)	2.60	0.48–17	0.5692
Age ^†^ (SD)	64.0 (5.72)	61.91 (5.90)	-	−5.30–9.48	0.5519
Rural origin	1 (25%)	1 (9.09%)	2.2	0.35–8.4	0.4762
FEV1/FVC ^†^ (SD)	46.88 (3.90)	43.39 (16.53)	-	−14.95–21.93	0.6892
FEV1% ^†^ (SD)	1.13 (0.35)	0.95 (0.50)	-	−0.42–0.78	0.5338
Biomass exposure	2 (50%)	6 (54.55%)	0.87	0.18–4.15	0.9999
Current smokers	4 (100%)	7 (63.64%)	0.63	0.35–1.34	0.5165
mMRC > 2	4 (100%)	11 (100%)	-	-	-
GOLD classification categories 1–2	2 (50%)	3 (27.27%)	0.75	0.27–1.45	0.5604
GOLD classification categories 3–4	2 (50%)	8 (72.73%)	2	0.41–8.93	0.5604
Comorbidities	3 (75%)	10 (90.91%)	1.53	0.69–8.26	0.4762
Microbiological isolation	1 (25%)	8 (72.73%)	1.77	0.90–4.82	0.2352
Lactate dehydrogenase (U/µL)	171.80 (30.96)	203.70 (60.05)	-	−101–37.05	0.303
C reactive Protein (mg/mL)	45.80 (67.60)	82.02 (141.5)	-	−226.20–153.80	0.6830
pO_2_ (mm Hg)	55 (11.30) ^•^	55.18 (20.38)	-	−33.55–33.19	0.9906
Saturation (%)	64.50 (24.75) ^•^	83 (16.60)	-	−48.10–11.10	0.1963

^†^ Median, COPD = chronic obstructive pulmonary disease, SD = standard deviation, CI = confidence interval, FEV1/FVC = the volume exhaled at the end of the first second of forced expiration/forced volume capacity ratio, FEV1% = percentage of volume exhaled at the end of the first second of forced expiration, mMRC = modified dyspnea scale of the British Medical Research Council, GOLD = Global Initiative for Chronic Obstructive Lung Disease, ^•^ = sample was only obtained from two patients, pO_2_ = oxygen pressure.

## Data Availability

The data presented in this study are available on request from the corresponding author. The data are not publicly available due to ethical and privacy reasons.

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
