# Peer review of "Pneumocystis jirovecii Colonization in Mexican Patients with Chronic Obstructive Pulmonary Disease"

_tropicalmed, 2023, doi:10.3390/tropicalmed8030137_

Round 1
Reviewer 1 Report
This is an interesting report of Pneumocystis prevalence in Mexico. No reports on colonization by Pneumocystis in Mexican COPD patients is available.
Unfortunately, the number of patients samples is too small to deserve major analyses. Indeed, Pneumocystis looks like improved the lung function of the sampled population as FEV1 looks better in the colonized patients (Table 2).
The title can be shortened to "Pneumocystis jirovecii colonization in Mexican patients with COPD".
The abstract and introduction suggest that the authors investigated the significance of colonization by this fungus. To study significance is required in light of the evidence from animal models. However, this would be a different study that would require of a large population of COPD patients and of strict and well defined inclusion and exclussion criteria. For example: smokers: refers to current smokers? or ex-smokers? These criteria needs better definition. Pack-year is missing,
The methods are well described, although can be shortened, especially since statistics in a population of 4 colonized versus 8 non-colonized subjects should be reconsidered.
Some sentences in the introduction and discussion sections and citations and comments need authors' data to support them. For example all the statements that refer to the effect of Pneumocystis in lung function, because the work shows only prevalence.
Nested PCR is an accepted procedure, therefore it can be used instead of real-time PCR. (See line 218).
The work presented is adequate evidence of Pneumocystis in COPD patients in Mexico and would support further studies to investigate the clinical significance of Pneumocystis in this population.
Author Response
Point 1: The title can be shortened to "Pneumocystis jirovecii colonization in Mexican patients with COPD".
Response 1: The title was changed accordingly. We use Chronic Obstructive Pulmonary Disease in place of COPD to avoid using abbreviations in the title
Point 2: The abstract and introduction suggest that the authors investigated the significance of colonization by this fungus. To study significance is required in light of the evidence from animal models. However, this would be a different study that would require of a large population of COPD patients and of strict and well defined inclusion and exclussion criteria. For example: smokers: refers to current smokers? or ex-smokers? These criteria needs better definition. Pack-year is missing
Response 2: We adjusted the text in the abstract and summary to specify the focus on prevalence in Mexican population. In our study, smokers were defined as current smokers, but we are aware of the lack of a well-defined definition (we lacked pack-year data for our sample of patients)
Point 3: The methods are well described, although can be shortened, especially since statistics in a population of 4 colonized versus 8 non-colonized subjects should be reconsidered
Response 3: We adjusted the methods to be more concise, and in considering an analysis of 4 colonized vs. 8 non-colonized patients, we consider that since we already have a statistical bias caused by the non-probabilistic selection of the sample (sampling bias), balancing the sample for the analysis of the groups would add a selection bias that could limit the validity of the results
Point 4: Some sentences in the introduction and discussion sections and citations and comments need authors' data to support them. For example all the statements that refer to the effect of Pneumocystis in lung function, because the work shows only prevalence
Response 4: We adjusted and added some citations to better sustain the sentences in the introduction and discussion and highlight the focus on prevalence in our paper.
Point 5: Nested PCR is an accepted procedure, therefore it can be used instead of real-time PCR. (See line 218).
Response 5: We eliminated the justification for the nested PCR used in this study.
Reviewer 2 Report
Lines 74-76 (Materials and Methods): “For the amplification of the genetic material of P. jirovecii, a 347 base pair (bp) fragment in the first reaction and a 260 bp fragment in the second reaction of the gene that codes for the mitochondria large subunit ribosomal ribonucleic acid gene (mt LSU rRNA) were the target sequences“ – here, the names of primers used in each step of nested-PCR should be mentioned.
Lines 94-98 (Materials and Methods): the part “To prevent contamination, pipettes with filters were used for all manipulations. DNA extraction and preparation of the reaction mixture were performed in 2 different rooms using separate laminar-flow hoods. The PCR procedure and analysis of PCR products were performed in another room. Control samples were performed simultaneously with the OPW samples.” – could be moved to the earlier paragraph, where positive and negative controls of the PCR are mentioned.
Lines 110-112 (Results): “All patients included in the study were without symptoms or signs of pneumonia at the time of the OPW sampling and were negative for HIV.” – these conditions are already mentioned in the exclusion criteria (Materials and Methods).
Lines 113-115 (Results): “The 347-bp fragment of the mt LSU rRNA gene of P. jirovecii was not amplified when using the samples (15 OPWs analyzed); however, 4 of the 15 samples were positive in the second round of amplification targeting the 260-bp fragment of the mt LSU rRNA gene by nested PCR.” – the structure of this sentence should be changed. It should be determined that none of the samples were positive in one-round PCR using primary reaction primers only, but 4 of them were positive when nested-PCR with secondary set of primers was applied (which met conditions for colonization definition in this study).
Lines 133-135 (Results): “Microbiological isolation of Pseudomonas aeruginosa, 1 (25%) patient with colonization vs. 8 (72.7%) patients without colonization, occurred in both groups (colonized and non- colonized).” – this sentence is unclear.
Line 143 (Discussion): “Our main findings and highlights are as follows.” – this sentence is unnecessary in my opinion. The introduction to discussion can be switched into a short summary of the most important effects of work, without listing them in sub-items.
The fact that OPW was the tested sample could be discussed more – another issue could be mentioned in the paragraph in lines 164-172: it has a lower sensitivity than for instance BAL, therefore the real prevalence may be in fact higher (as Pneumocystis colonization is characterized by low fungal burden, therefore it may be overlooked in less invasive type of biological sample). On the other hand, as the Authors mentioned, OPW is one of the non-invasive methods which have the advantage that the method of their collection is less harmful to the patient (and sometimes it is even impossible to collect e.g. BAL in patients in a serious health condition)
Lines 177-183 (Discussion): in my opinion, it is not necessary to justify the use of nested-PCR. Likewise, the point (III) from the Limitations of the study paragraph can be removed.
I suggest that Table 1 could be joined with Table 2 and even with Table 3. Besides, the headline of Table 3 is incorrect.
The short form “P. jirovecii” can be used instead of “Pneumocystis jirovecii” (for instance in Abstract, line 27). Similarly – COPD (e.g. line 35, Abstract).
The term “study group”/”group” should be used instead of “sample” (it occurs repeatedly in the text, for instance line 35 of Abstract).
“Pneumocystis” should always be in italics (for instance line 162).
Some sentences are too long and should be divided into two/three separate (for instance: lines 184-191; 195-198; 204-210).
Author Response
Point 1: Lines 74-76 (Materials and Methods): “For the amplification of the genetic material of P. jirovecii, a 347 base pair (bp) fragment in the first reaction and a 260 bp fragment in the second reaction of the gene that codes for the mitochondria large subunit ribosomal ribonucleic acid gene (mt LSU rRNA) were the target sequences“ – here, the names of primers used in each step of nested-PCR should be mentioned.
Response 1: We included the primer sequence of the primers used.
Point 2: Lines 94-98 (Materials and Methods): the part “To prevent contamination, pipettes with filters were used for all manipulations. DNA extraction and preparation of the reaction mixture were performed in 2 different rooms using separate laminar-flow hoods. The PCR procedure and analysis of PCR products were performed in another room. Control samples were performed simultaneously with the OPW samples.” – could be moved to the earlier paragraph, where positive and negative controls of the PCR are mentioned.
Response 2: We moved this part to the previous paragraph.
Point 3: Lines 110-112 (Results): “All patients included in the study were without symptoms or signs of pneumonia at the time of the OPW sampling and were negative for HIV.” – these conditions are already mentioned in the exclusion criteria (Materials and Methods).
Response 3: We deleted this part to reduce redundancy.
Point 4: Lines 113-115 (Results): “The 347-bp fragment of the mt LSU rRNA gene of P. jirovecii was not amplified when using the samples (15 OPWs analyzed); however, 4 of the 15 samples were positive in the second round of amplification targeting the 260-bp fragment of the mt LSU rRNA gene by nested PCR.” – the structure of this sentence should be changed. It should be determined that none of the samples were positive in one-round PCR using primary reaction primers only, but 4 of them were positive when nested-PCR with secondary set of primers was applied (which met conditions for colonization definition in this study).
Response 4: We changed it accordingly for clarity
Point 5: Lines 133-135 (Results): “Microbiological isolation of Pseudomonas aeruginosa, 1 (25%) patient with colonization vs. 8 (72.7%) patients without colonization, occurred in both groups (colonized and non- colonized).” – this sentence is unclear.
Response 5: The text was adjusted for clarity.
Point 6: Line 143 (Discussion): “Our main findings and highlights are as follows.” – this sentence is unnecessary in my opinion. The introduction to discussion can be switched into a short summary of the most important effects of work, without listing them in sub-items.
Response 6: We adjusted this paragraph accordingly and eliminated the first sentence.
Point 7: The fact that OPW was the tested sample could be discussed more – another issue could be mentioned in the paragraph in lines 164-172: it has a lower sensitivity than for instance BAL, therefore the real prevalence may be in fact higher (as Pneumocystis colonization is characterized by low fungal burden, therefore it may be overlooked in less invasive type of biological sample). On the other hand, as the Authors mentioned, OPW is one of the non-invasive methods which have the advantage that the method of their collection is less harmful to the patient (and sometimes it is even impossible to collect e.g. BAL in patients in a serious health condition).
Response 7: We added some discussion for the OPW sampling, justification, and comparison with BAL, and also added 2 new references to the text in this paragraph.
Point 8: Lines 177-183 (Discussion): in my opinion, it is not necessary to justify the use of nested-PCR. Likewise, the point (III) from the Limitations of the study paragraph can be removed
Response 8: We eliminated the justification and limitations.
Point 9: I suggest that Table 1 could be joined with Table 2 and even with Table 3. Besides, the headline of Table 3 is incorrect.
Response 9: We combined the three tables and changed the text and titles accordingly.
Point 10: The short form “P. jirovecii” can be used instead of “Pneumocystis jirovecii” (for instance in Abstract, line 27). Similarly – COPD (e.g. line 35, Abstract).
Response 10: We made the suggested change.
Point 11: The term “study group”/”group” should be used instead of “sample” (it occurs repeatedly in the text, for instance line 35 of Abstract).
Response 11: We made the suggested change.
Point 12: “Pneumocystis” should always be in italics (for instance line 162)
Response 12: We made the suggested change.
Point 13: Some sentences are too long and should be divided into two/three separate (for instance: lines 184-191; 195-198; 204-210).
Response 13: We made the suggested changes.
Round 2
Reviewer 1 Report
The manuscript can be publshed in its present form after English is evaluated by a native speaker.
Author Response
The paper has undergone language editing by MDPI, we uploaded the editing certificate. Additionally, we made the modifications suggested by reviewer 2 (this minor modifications to the abstract were only send to us via e-mail) and can be seen in the final version of the manuscript.
